# Improved Protocol for Efficient *Agrobacterium*-Mediated Transient Gene Expression in *Medicago sativa* L.

**DOI:** 10.3390/plants13212992

**Published:** 2024-10-26

**Authors:** Suma Basak, Dipika Parajulee, Seema Dhir, Ankush Sangra, Sarwan K. Dhir

**Affiliations:** 1Center for Biotechnology, Department of Agricultural Sciences, Fort Valley State University, Fort Valley, GA 31030, USA; dparajul@wildcat.fvsu.edu (D.P.); dhirs0@fvsu.edu (S.K.D.); 2Department of Biology, College of Arts and Sciences, Fort Valley State University, Fort Valley, GA 31030, USA; dhirs@fvsu.edu; 3Department of Genetics, University of Georgia, Athens, GA 30602, USA; ankush.sangra@fvsu.edu

**Keywords:** *Medicago sativa* L. (alfalfa), *Agrobacterium tumefaciens*, *β-glucuronidase* (GUS), green fluorescent protein (GFP), transient gene

## Abstract

*Medicago sativa* L. (Alfalfa) is a globally recognized forage legume that has recently gained attention for its high protein content, making it suitable for both human and animal consumption. However, due to its perennial nature and autotetraploid genetics, conventional plant breeding requires a longer timeframe compared to other crops. Therefore, genetic engineering offers a faster route for trait modification and improvement. Here, we describe a protocol for achieving efficient transient gene expression in alfalfa through genetic transformation with the *Agrobacterium tumefaciens* pCAMBIA1304 vector. This vector contains the reporter genes β-glucuronidase (GUS) and green fluorescent protein (GFP), along with a selectable hygromycin B phosphotransferase gene, all driven by the CaMV 35s promoter. Various transformation parameters—such as different explant types, leaf ages, leaf sizes, wounding types, bacterial concentrations (OD_600nm_), tissue preculture periods, infection periods, co-cultivation periods, and different concentrations of acetosyringone, silver nitrate, and calcium chloride—were optimized using 3-week-old in vitro-grown plantlets. Results were attained from data based on the semi-quantitative observation of the percentage and number of GUS spots on different days of agro-infection in alfalfa explants. The highest percentage of GUS positivity (76.2%) was observed in 3-week-old, scalpel-wounded, segmented alfalfa leaf explants after 3 days of agro-infection at a bacterial concentration of 0.6, with 2 days of preculture, 30 min of co-cultivation, and the addition of 150 µM acetosyringone, 4 mM calcium chloride, and 75 µM silver nitrate. The transient expression of genes of interest was confirmed via histochemical GUS and GFP assays. The results based on transient reporter gene expression suggest that various factors influence T-DNA delivery in the *Agrobacterium*-mediated transformation of alfalfa. The improved protocol can be used in stable transformation techniques for alfalfa.

## 1. Introduction

*Medicago sativa* L. (alfalfa) is a popular forage legume crop to produce high-quality biomass that is utilized for hay, silage, cover crops, and green manure [1]. The cultivation of alfalfa is valued for its role in soil protection, bioremediation, the nitrogen cycle, and wildlife habitat preservation. Being a perennial crop with nitrogen-fixing capabilities, it is extensively cultivated as a primary source of proteins, enzymes (amylase, coagulase, peroxidase, pepsin, lipase, invertase, and pectinase), antioxidants, minerals, and vitamins A, C, K, and E, as well as valuable phytopharmaceutical components [2]. Moreover, it can be used as a cover crop in grasslands for improved weed control, in the production of recombinant pharmaceutical proteins [3], and in phytoremediation [4]. Alfalfa is an autotetraploid with a chromosome count of 32 (2n = 4x = 32) and a genome size ranging from 800 to 900 Mbp in length [5]. Researchers have undergone more than 50 years of efforts aimed at enhancing the productivity and quality traits of alfalfa. However, success has been limited [6], due to the intricacy of this species and the challenges involved in its genetic enhancement using traditional breeding methods. The utilization of biotechnological tools and techniques is essential to facilitate the large-scale production of alfalfa for industrial use.

Genetic transformation introduces novel traits or manipulates gene expression, leading to valuable phenotypic variations that are crucial for crop enhancement and to improve the understanding of gene function. This process entails integrating exogenous genes and regulating endogenous gene expression through transformation [7]. Various techniques are available for transferring desired genes into the plant genome, via direct and indirect techniques such as electroporation, microprojectile bombardment, and *Agrobacterium*-mediated transformation [8]. Among these, the *Agrobacterium*-mediated technique has become a common practice in numerous laboratories [9], involving the transfer of DNA from the *Agrobacterium* plasmid into the cells of numerous dicotyledonous and some monocotyledonous plants [10]. Since the initial report by Deak et al. (1986) [11], multiple efforts have been made to introduce beneficial genes into the alfalfa genome via *Agrobacterium*-mediated transformation, such as insect resistance against *H. postica* larvae [12], increased salt tolerance [13,14], improved nutritional quality [15], higher yields via delayed leaf senescence [16], and higher rates of phosphate uptake into cells [17].

Reporter genes utilized as visual markers allow for the visualization of gene expression and protein localization in vivo for a wide range of prokaryote and eukaryote genomes. For example, genes encoding chloramphenicol acetyltransferase (CAT), green fluorescent protein (GFP), luciferase (LUC), β-glucuronidase (GUS), and red fluorescent protein (DsRed2) are commonly employed as reporters [18,19]. The *Escherichia coli* gusA (*uidA*) gene encoding for the enzyme β-glucuronidase (GUS) offers a clear indication that transferred genes are expressed in transgenic plant cells in either a transient or stable manner [20,21]. The GUS gene cleaves a β-glucuronidase substrate variant, which is used in spectrophotometric, fluorometric, and histochemical assays, resulting in the formation of an indigo–blue precipitate within cells. Green fluorescent protein (GFP), isolated from jellyfish (*Aequorea victoria*), is a powerful tool in molecular biology and cell biology research. It serves as a widely utilized reporter gene across numerous living organisms [22,23], frequently aiding in the identification of transformed cells, tissues, and organs in both plants and animals, often along with genes conferring antibiotic or herbicide resistance. GFP’s strong fluorescent intensity makes it an excellent visual marker for selecting positive transformants and has the potential to track the expression of transgenes in genetically engineered crops. In addition, it does not require any exogenous substrates. This also allows for the monitoring of transgenic expression from early events of transformation through the regeneration of transgenic plants. Another advantage is the relatively small (26.9 kD) genome size, which assists with both N- and C-terminal protein fusions, lending to the study of localization and intracellular protein trafficking [23].

Genetic engineering technology has been used in alfalfa breeding programs to target the improvement of interesting agronomic traits such as herbicide tolerance and forage quality. Roundup Ready alfalfa [24] and low-lignin alfalfa have been deregulated and commercialized in the US. Various methods have been attempted for the genetic transformation of alfalfa [11,25,26,27]. Since the identification of a highly regenerable genotype, Regen SY4D [28,29], *Agrobacterium*-mediated transformation of alfalfa has become easier and more efficient. To date, the transformation method that uses infection of the leaf as an explant by *Agrobacterium tumefaciens,* followed by the production of somatic embryos for transgenic plant regeneration, has been widely used in many laboratories [26]. Improving the efficiency of *Agrobacterium*-mediated transformation is crucial for enhancing virulence and increasing transformation frequency. Despite scattered studies on limited genotypes of alfalfa for transformation, the optimization and establishment of these protocols have primarily focused on limited parameters such as the use of different explants, genotypes, or cultivars; hormone concentrations and ratios in plant regeneration media; in planta transformation; selection schemes; and *Agrobacterium* strains and transformation vectors [9,30,31,32,33,34,35]. However, many of these methods are either complex and costly or yield lower transient and stable transformation frequency and subsequent plant regenerations. A review of the existing literature has made it apparent that replicating the findings of prior studies presents significant challenges for the large-scale production of alfalfa for the pharmaceutical industry. These challenges occur within various genotypes where a large amount of variation is present. This requires the optimization of efficient transformation protocols that can be applicable to commercial genotypes.

Transient expression is a critical step for successful transformation, which is widely utilized across various areas such as gene expression, functional studies, plant biological material production [8], promoter characterization [36], gene silencing [37,38], elicitor identification [39], vaccine production [40], and CRISPR/Cas9-based genome editing [38,41]. Despite its widespread utility, a comprehensive review of the literature indicates that the optimization of transient gene expression in alfalfa remains underexplored and has not been directly addressed in extensive transformation studies. Previous research in this area has predominantly used single reporter genes and relied on kanamycin as a selectable marker [26,27], overlooking the superior efficacy of hygromycin in distinguishing transformed from non-transformed cells. To address this gap, the present study aims to establish a robust, versatile, and systematic protocol for optimizing *Agrobacterium*-mediated transient gene expression in alfalfa. Our research incorporates dual reporter genes, GUS and GFP, with hygromycin as a selectable marker, thereby enhancing the efficiency and effectiveness of transformation protocols in alfalfa (Figure 1). Therefore, the main objective of this study is to evaluate and improve various parameters in alfalfa genotypes to develop a highly efficient transformation system. This transient gene expression system aims to enhance the production of transgenic plants and, in the long-term, advance gene-editing techniques in alfalfa.

## 2. Results

Transient assays are a valuable resource for examining gene function in plants. The *Agrobacterium*-*mediated* transformation approach is a rapid and exceptionally efficient technique that does not require costly equipment. Multiple methods have been documented to enhance transformation effectiveness for different plant species, as demonstrated in [42]. In this study, the effectiveness of the *Agrobacterium*-mediated transformation system in *M. sativa* was assessed by observing the influence of various factors using pCAMBIA1304 vector (Figure 1). The success of the transformation was determined by the percentage of observed blue spots, indicating transient expression of GUS-positive transformants. GFP gene expression was identified by the presence of green fluorescent cells.

### 2.1. Effects of Explant Types, Leaf Age, and Sizes of Explants on Transient GUS Expression

The type of explant used influences the ability of the *Agrobacterium* to transfer T-DNA into the genome of the host plant cell [43,44,45,46]. Five different explants from healthy 3-week-old subcultured alfalfa plant—leaf, stem, petiole, root, and leaf-derived embryogenic callus, initiated at 3 weeks in the dark—were selected as the target explants for this transient gene expression study. Approximately 2 mm leaf segments were cultured in a callus induction medium. Callus initiation started with the curling of leaf layers, and after one week, a greenish or yellowish callus was observed. The different phases of callus formation are presented in Figure 2A–C. To optimize the transient expression protocol, the transformation efficiencies of different explants were evaluated (Figure 3A–H). Our study revealed variations in transformation frequencies, with the highest transient GUS expression observed in 76.2% of leaf explants, followed by 38.7% in stems, 20.3% in petioles, 14.5% in roots, and 42.9% in 3-week-old leaf-derived callus of alfalfa (Figure 3A–H and Figure 4A).

The age of the leaf explant is a critical determinant of transient GUS expression [47]. The transient GUS expression decreased with the increasing age of the leaf explants. Young leaves, specifically those subcultured at 3-weeks of age, exhibited a markedly higher transient GUS expression (74.2%) compared to older leaves at 5 and 7 weeks, which only exhibited 56.6% and 48.4% expression, respectively (Figure 4B). Leaves that were less than one week old yielded a transient GUS expression of only 30.1%. Conversely, leaf explants aged 9 weeks exhibited higher resistance to agro-infection, with only 28.6% GUS expression observed. One possible explanation could be that the older leaves possess a more robust defense mechanism, potentially diminishing the ability of *Agrobacterium* to grow and transfer T-DNA. However, leaves aged more than 5 weeks are too brittle for agro-infection. The sizes of leaf explants are also a crucial factor affecting transient GUS expression. Transformation efficiency was significantly decreased in whole or intact leaf explants compared to segmented ones. The segmented leaf explants demonstrated the highest GUS expression, at 91.1%, whereas only 56.2% expression was observed in the whole explants (Figure 3A,D and Figure 4C).

### 2.2. Effect of Wounding Types on Transient GUS Expression

In nature, *Agrobacterium* infects plants through natural openings, such as wounds. Therefore, the wounding or pricking of the plant cells is essential to establish a pathway to *Agrobacterium* infection, enabling the delivery of T-DNA to explants [48]. This is required to overcome physical barriers such as waxy cuticles on the plant’s epidermis, which could impede T-DNA transfer [47]. Additionally, a plant’s wound response stimulates the release of phenolic compounds for the activation of *vir* genes. Furthermore, cell division is stimulated, and the cell walls become less rigid, making it easier for the bacterium to breach. At this stage, active DNA replication can assist in T-DNA integration [48]. In this study, the effects of different wounding types of leaf explants were examined. There was a significant increase (over 70%) in T-DNA transfer observed in wounded plants as compared to unwounded explants infected with *Agrobacterium*. The highest transient GUS expression was observed with scalpel wounding at 86.3%, followed by needle wounding at 53.5%, mash wounding at 49.6%, cell strainer pestle wounding at 35.9%, and the control or unwounded or intact at 12.9% (Figure 5).

### 2.3. Effects of Agrobacterium Concentrations or Optimal Density (OD_600nm_) on Transient GUS Expression

The efficiency of transformation depends upon the concentration of *Agrobacterium*, as adequate attachment of *Agrobacterium* to plant cells is crucial for successful gene transfer to take place [49]. Different plants exhibit varying transformation efficiencies with *Agrobacterium* concentrations ranging from 0.1 to 2.0 [50]. In this study, we tested the effect of *Agrobacterium* concentration on transformation efficiencies and found that the specific density (0.1–1.2 OD*_600nm_*) of *Agrobacterium* resulted in distinct levels of transient GUS expression in alfalfa explants. After 3 days of co-cultivation, the transformation rate of alfalfa explants significantly increased, with the *Agrobacterium* concentration ranging from 0.1 to 0.6 OD*_600nm_*. In this study, the greatest percentage of GUS-stained alfalfa explants (63.6%) was observed in a suspension culture of the *Agrobacterium*, at 0.6 OD*_600nm_*, followed by 0.4 and 0.8 with 55.9% and 45.3% GUS expression, respectively (Figure 6). However, the results obtained at 0.6 OD*_600nm_* were not significantly different from those at 0.4 OD*_600nm_*. High concentrations of *Agrobacterium* suspension cultures (1.0 and 1.2 of OD*_600nm_*) significantly decreased the number of transformed cells by 42.5% and 30.5%, respectively. High *Agrobacterium* concentrations increased the number of infected plant cells, but they also disrupted the physiological processes in the explants, leading to the termination of the transformation process.

### 2.4. Effects of Preculture Periods on Transient GUS Expression

A preculture period involves the culturing of the explants in preculture medium for different durations before *Agrobacterium* infection and co-cultivation [51]. During this time, the plant tissues undergo physiological and developmental changes that significantly enhance their ability to regenerate, thereby promoting transformation efficiency [45]. As the preculture medium supplemented with growth regulators generally promotes cell division, actively dividing cells are more suitable for the T-DNA delivery and integration of transgenes [52]. The impacts of varying preculture periods of leaf tissues (ranging from 1 to 7 days) on the efficiency of alfalfa’s transient expression were evaluated. The results showed that alfalfa segmented leaf precultured for 2 days yielded the highest positive GUS results, at 72.7%, followed by preculture periods of 1 and 3 days, at 51.9% and 44.5%, respectively (Figure 7). With longer preculture periods (4, 5, 6, and 7 days), the transformation efficiency decreased to 43.8%, 38.7%, 29.6%, and 19.1%, respectively. The extended preculture period allowed for the alfalfa explants to increase their nutrient uptake from the medium, stimulating cells to become competent for *Agrobacterium* attachment.

### 2.5. Effects of Infection Periods on Transient GUS Expression

To achieve efficient *Agrobacterium* attachment and subsequent transformation of plant cells, a suitable infection period is required. Precultured alfalfa leaf explants were immersed in an *Agrobacterium* suspension culture with an OD*_600nm_* of 0.6 for varying periods of 5 to 35 min before co-cultivation. The duration of immersion significantly affected GUS expression, with the highest transient expression (78.5%) observed after 30 min, followed by 25 and 35 min, which displayed GUS expression levels of 72.3 and 62.9%, respectively (Figure 8). Infection duration of less than 25 min resulted in lower percentages of 30.9, 39.5, 50.3, and 56.3% GUS expression for 5, 10, 15, and 20 min, respectively, due to insufficient time for the *Agrobacterium* to infect the leaf explants. However, extending the infection period beyond 35 min decreased the transient GUS expression due to *Agrobacterium* overgrowth, leading to stress and subsequent browning or blackening of the explants. Such stress could trigger necrotic and hypersensitive symptoms and negatively affect transformation rates.

### 2.6. Effects of the Co-Cultivation Periods on Transient GUS Expression

Co-cultivation periods allow for T-DNA expression in cells and provide time for the recovery of transformed cells to express the introduced DNA. However, in different studies, it is also meant for the post-incubation period, which refers to the time between the removal of unattached *Agrobacterium* after agro-infection and the observation or selection of transgenic lines from the explants [51,53,54,55,56]. During this stage, alfalfa leaf explants are cultured with attached *Agrobacterium* to facilitate further infection on a medium lacking selection pressure and anti-*Agrobacterium* agents. The transient expression of GUS has been observed in alfalfa leaf explants co-cultivated with *Agrobacterium* for different periods (1 to 7 days). The highest frequency of 83.5% was achieved after 3 days of co-cultivation, while the transformation frequencies were much lower for 1- and 2-day co-cultivation periods, at 35.2% and 45.7%, respectively (Figure 9). Co-cultivation for 2–3 days has been used in *Spinacia oleracea* L. [45] and *Moringa oleifera* Lam. [57]. Co-cultivation periods of less than 3 days produced few transformed lines, due to insufficient time for complete or maximized transfer of *Agrobacterium* T-DNA into the alfalfa genome. Liu et al. (2008) [58] reported that no transient GUS-positive explants and spots in soybeans were found within 0 days of a co-cultivation period. After 4 days of co-cultivation in alfalfa leaf explants, around 62.5% transient expression was observed, and for a co-cultivation period of 5, 6, and 7 days, the frequency of blue spots on the leaf explants was 42.9%, 28.9%, and 16.7%, respectively.

### 2.7. Effects of Acetosyringone Concentrations on Transient GUS Expression

Acetosyringone (AS), a phenolic inducer that affects the virulence genes of *Agrobacterium*, is essential for infection. AS is also necessary for the infection and co-cultivation period to facilitate the transfer and incorporation of T-DNA into the recalcitrant host plant—particularly monocotyledons, which cannot naturally synthesize this compound and, therefore, are not suitable hosts for *Agrobacterium*. Alfalfa leaf explants were immersed in different concentrations of AS, ranging from 50 to 350 µM, in order to assess its effect on transformation. The results showed that the transient transformation efficiency significantly improved up to a concentration of 150 µM, indicating the critical role of AS in alfalfa transformation (Figure 10). In species where low levels of AS were present, such as *Oncidium* and *Cymbidium* orchids, *Carrizo,* and *Hamlin* lime, the *vir* genes were induced synergistically by sugars, e.g., glucose and galactose, leading to an increase in transformation efficiency [59]. In the case of alfalfa, the maximum percentage (71.4%) of GUS-positive leaf explants was observed with the addition of 150 µM AS, followed by 46.8% in 100 µM and 39.1% in 200 µM AS. However, at concentrations higher than 150 µM (250, 300, and 350 µM produced GUS expression of 33.2, 25, and 11.1%, respectively), the transient transformation efficiency decreased, and in some cases, it was almost completely inhibited. A similar result was obtained in the transformation of *Dendrobium* orchids, where higher concentrations of AS resulted in the inhibition of transformation. The inhibition at higher concentrations could be attributed to the toxicity of AS to explants, which produced hypersensitive symptoms due to the harmful effect of supra-optimal AS concentration on *Agrobacterium* inoculation, thus affecting the transformation rate [20].

### 2.8. Effects of Calcium Chloride (CaCl_2_) Concentrations on Transient GUS Expression

It is known that calcium is an essential macronutrient for plants, playing a crucial role in the cell wall structure; it acts as an ionic cross-linkage, binding carboxyl groups of linear macromolecules in the plant cell wall. Typically, the plant cell wall contains a concentration of 1–10 mM calcium chloride (CaCl_2_), which can make up more than 60% of the total calcium in the cell [20,60,61]. Research has shown that an increase in CaCl_2_ concentration can stimulate cell dissociation, leading to hyper-secretion of polysaccharide compounds [62]. Bacteria were made competent with CaCl_2_ and transformed with plasmids by heat shock [63]. In this study on alfalfa leaf explants, different treatments with varying CaCl_2_ strengths (ranging from 1 to 10 mM) were evaluated, and the highest percentage of GUS expression (68.4%), a marker for transient gene expression, was observed at a CaCl_2_ concentration of 4 mM, followed by 58.2% and 47.3% for CaCl_2_ concentrations of 3 and 5 mM, respectively (Figure 11). However, transient GUS activity decreased by 23.8% when the CaCl_2_ concentration exceeded 6 mM.

### 2.9. Effects of Silver Nitrate (AgNO_3_) Concentrations on Transient GUS Expression

Ethylene, a colorless hydrocarbon-based gas, is produced during agro-infection, which can reduce the efficiency of gene transfer mechanisms in *Agrobacterium*-mediated transformation. However, silver nitrate (AgNO_3_) has been found to inhibit ethylene production, which affects cell division mechanisms [64]. Ethylene production is enhanced by wounding during explant preparation, which can lead to browning or tissue damage. Nevertheless, adding AgNO_3_ to the co-cultivation medium at an optimal level can significantly suppress *Agrobacterium* growth, without compromising T-DNA delivery and subsequent T-DNA integration, by reducing the capacity of Ag^2+^ to bind to the ethylene receptor produced through ethylene biosynthesis [65].

Tissue browning is a significant factor that can reduce transformation efficiency in plants [66], including alfalfa. To assess transformation efficiency and prevent tissue browning based on transient GUS expression, the impacts of varying concentrations of AgNO_3_ (10–150 µM) on alfalfa leaf explants were investigated. The results showed that the highest percentage of transient GUS gene expression observed at 75 µM (69.1%), followed by 38.3% and 46.5% at 50 µM and 100 µM, respectively. In contrast, 33.6% expression was observed at 150 µM AgNO3 (Figure 12). We postulate that this could be attributed to the phytotoxic effects of high AgNO_3_ concentrations during co-cultivation on alfalfa leaf explants.

### 2.10. Green Fluorescence Protein (GFP) Gene Transient Expression

Successful gene delivery into alfalfa explants was confirmed through GFP gene expression, with transformed cells displaying GFP production and becoming distinguishable from non-transformed cells. Transient green fluorescence was detected in alfalfa leaf explants (Figure 13A–F) and leaf-derived calli (Figure 13G–L) compared with non-transformed plants, which emitted chlorophyll-induced red autofluorescence. The leaves of alfalfa explants produced GFP after 3 days of co-cultivation with *Agrobacterium,* while the GFP-positive callus was developed 3-weeks after co-cultivation in an optimized medium, and then gradually decreased in intensity thereafter. Approximately 58% of the alfalfa explants exhibited transient GFP gene expression. Notably, severe tissue necrosis, characterized by browning of the culture, was observed after co-cultivation period. This was likely due to callus growth on Gamborg’s B5 basal (B5H) medium, supplemented with 3% maltose, 4.5 μM 2,4-dichlorophenoxyacetic acid (2,4-D), 0.9 μM kinetin, 6.65 g of glutamine, 0.8 g of serine, 0.004 g of adenine, and 0.08 g of L-glutathione, 400 mg/L cefotaxime, and 12.5 mg/L hygromycin. GFP-positive callus formations were detected from co-cultivated explants after 28 days of culture (data not shown). The expression levels of the two reporter genes, GUS and GFP, were consistent in explant used for transient gene expression.

### 2.11. Molecular Analysis on Transgenic Tissue

The selected explants were transferred to Gamborg’s B5 basal (B5H) medium, supplemented with 3% maltose, 4.5 μM 2,4-dichlorophenoxyacetic acid (2,4-D), 0.9 μM kinetin, 6.65 g/L glutamine, 0.8 g/L serine, 0.004 g/L adenine, 0.08 g/L L-glutathione, 400 mg/L cefotaxime, and 12.5 mg/L hygromycin. After 3 weeks, callus formation was initiated, followed by somatic embryo development. The putative transgenic plants were regenerated from the matured embryos. To confirm the presence of stable GUS gene expression in alfalfa putative transformant explants (leaf, stem, petiole, root, and 3-week-old leaf-derived callus), polymerase chain reaction (PCR) amplification was performed on their genomic DNA. Genomic DNAs of alfalfa were extracted from both putative transformed and non-transformed explants. Using GUS-specific primers, a PCR product of the expected 493 bp fragment was detected in all transformed explants (Lanes 2–6), which tested positive for GUS in the GUS assay, while no amplification was observed in non-transformed explants (Lanes 7–11) (Figure 14). The detection of the 493 bp GUS fragment by PCR aligned with reference genes, confirming the presence of the GUS gene through transient GUS gene expression and reducing the likelihood of false positives.

## 3. Discussion

Tissue culture and transformation are fundamental techniques in functional plant genomics. Although *Agrobacterium*-mediated transformation has been widely employed in various plant species for decades, some legume crops, such as alfalfa, continue to pose challenges for achieving efficient genetic transformation and high-level expression of target genes [9]. Since the first report of *Agrobacterium tumefaciens*-mediated transformation in alfalfa [11], numerous studies have investigated various methods for the regeneration and genetic transformation of alfalfa [9,32,33], although these methods often prove less effective for this species. Identifying all factors that affect transformation efficiency is crucial for developing effective expression systems in recalcitrant legume crops such as alfalfa. *Agrobacterium*-mediated transformation for transient gene expression is particularly useful for this purpose, as it facilitates the rapid testing of a variety of parameters. In comparison to existing transformation procedures, our newly established method offers several advantages. Our research on transient expression in alfalfa has shown that different explants can successfully express GUS and GFP genes. This study highlights that our transient expression protocol effectively optimizes key parameters, leading to high levels of GFP and GUS expression.

The types of explants, age of leaves, and tissue sizes are crucial factors in *Agrobacterium*-mediated transient GUS expression. In this study, we found that leaf explants excised from 3-week-old plants exhibited the highest GUS expression levels compared to stem-, petiole-, root-, and leaf-derived embryogenic calli. One possible explanation for this could be that intact or whole leaves older than 3 weeks have a more developed defense mechanism, which might reduce the ability of *Agrobacterium* to grow and transfer T-DNA. In addition, the highest transient GUS expression was found in wounding with a scalpel compared to other approaches. Several methods of wounding are employed in plant transformation, including small incisions made with blades, micro-wounding using particle guns or sonication, and injection via a syringe [49]. However, mechanical wounding has significantly improved transformation efficiency in some plant species, including recalcitrant grain legumes such as green gram [52], black gram [67], soybeans [48], and chickpeas [51].

The optimal density of the *Agrobacterium* culture significantly influences both stable and transient transformation efficiency. Our finding of higher GUS expression with the *Agrobacterium* concentration at 0.6 OD*_600nm_* is consistent with those of previous studies conducted in chickpeas [51,56], rice [53], ramie [68], peanuts [69,70], and soybeans [48]. The results highlight that both variable *Agrobacterium* concentrations and plant species are crucial factors in the efficacy of plant transformation mediated by *Agrobacterium.* Our findings also highlight the importance of optimizing the preculture period for successful plant transformation experiments, as the transient gene expression in our study had high transformation efficiency at 2 days of preculture. While successful applications of explant preculture before *Agrobacterium* infection have been observed in various leguminous plants, such as soybeans [71], peanuts [69], green gram [72], and chickpeas [51], it is nevertheless crucial to note that the ideal preculture period may differ for other plant species [57] and can be influenced by numerous factors, such as the type of tissue and the transformation method employed.

Notably, the optimal infection period is variable among plant species and explant tissue types. For instance, a 30 min infection period was sufficient for transient expression in cotyledonary nodes of soybeans [44] and chickpeas [46], while a 60 min infection period was needed for in vitro *Melastomataceae* spp. explants [73]. In our study on infection duration, the highest transient expression (78.5%) was observed after 30 min. During the co-cultivation process, explants were incubated with *Agrobacterium*. The transient GUS expression was low after 1 day of co-cultivation, and the maximum result was at 3 days in our study, which is consistent with previous findings reported in *Dierama erectum* Hilliard [49], broccoli [74], bananas [66], chickpeas [46], and *Phalaenopsis violacea* orchids [20]. However, longer co-cultivation periods of 6–7 days also significantly reduced the transient GUS expression efficiency due to the overgrowth of other bacteria, with a contaminant resulting in suffocation of the alfalfa leaf explants and turning necrotic. Comparable results have been reported in chickpeas [55], *Vigna radiata* [72], *Vigna mungo* [75], and *Glycine max* [48].

Li et al. (2022) [76] demonstrated the involvement of various molecules in the transfer of T-DNA from bacterium to host, stimulated by proteins encoded by the virulence (*vir*) genes. Acetosyringone (AS) is one of these phenolic inducer molecules that stimulate the genes of *Agrobacterium*. However, in some plant species—such as poplar, Mexican lime, and tobacco—AS did not improve the transformation efficiency in the *Agrobacterium*-mediated system, possibly due to the production of high quantities of phenolics secreted by the plant tissues, rendering AS unable to induce virulence and infection [59]. Nevertheless, the addition of AS during co-cultivation and infection has been shown to significantly improve the *Agrobacterium*-*mediated* system in some plants, including *Dierama erectum* Hilliard [49], *Phalaenopsis* orchids [20], and *Spinacia oleracea* L. [45]. In our study, the transient transformation efficiency significantly improved at a concentration of 150 µM, which is comparable to previous findings where the application of 100–500 µM AS to the co-culture medium was effective for *Agrobacterium*-mediated transformation in peanuts [70], black gram [75], cowpeas [77], green gram [72], soybeans [48], and chickpeas [51].

Calcium is a vital macronutrient that is essential for cell wall repair. Reduced calcium concentrations can cause cell wall deterioration. This study demonstrated that supplementation with 4 mM calcium chloride enhanced transient transformation efficiency. Interestingly, a lack of calcium in plants may increase the transformation rate, as it can alter the structure of the cell wall, reducing the matrix and making it less conducive to *Agrobacterium* cell attachment, as supported by the results of transient GUS gene expression in the calcium-free medium [78,79]. On the other hand, higher calcium concentrations can reinforce the plant cell wall by being absorbed into it. Calcium also acts as an inducer, influencing gene expression and changing the composition of the cell wall. Strengthening the cell wall may result in lower rates of T-DNA gene segment transfer from *Agrobacterium* to plant cells.

Silver nitrate (AgNO_3_) is an anti-necrotic agent that can minimize oxidative bursts during the interaction between plant tissue and *A. tumefaciens* [66,80,81]. The addition of AgNO_3_ to the co-culture medium has been known to exhibit anti-ethylene activity, which is a common occurrence during in vitro plant cultures. This subsequently slows down *Agrobacterium* growth on the target explants, ultimately facilitating plant cell recovery and increasing transformation efficiency in various plant species, such as soybeans [82], cassava [83], *Prunus avium* (L.) cv Stella [80], maize [84], apple [85], *Phalaenopsis violacea*, and *Dendrobium* orchids [20]. The use of AgNO_3_ has been found to have significant effects on plant tissue culture, including enhancement of somatic embryogenesis, organogenesis, and micropropagation in many species [86]. In the present study, the maximum transient transformation efficiency improved at a concentration of 75 µM. Previous studies have shown that the addition of varying concentrations of AgNO_3_ increased the rate of shoot elongation in the 88-1 soybean variety. Moreover, the transformation efficiency of the 88-1 soybean variety also showed an improvement, rising from 3.2 to 5.5% upon the addition of 15 mg/L AgNO_3_ [44]. In addition, AgNO_3_ has been shown to stimulate direct shoot regeneration in in vitro cultures and, when co-cultivated with *A. tumefaciens* for genetic transformation studies, it inhibits bacterial growth after co-cultivation. For instance, the addition of 100 mg/L AgNO_3_ to the plant regeneration medium after co-cultivation can completely halt bacterial growth for at least 3 weeks. As a result, AgNO_3_ can serve as a substitute for commonly used antibiotics such as cefotaxime and carbenicillin, which are frequently employed during the initial passage after co-cultivation with *Agrobacterium* strains in wheat [61,87].

In our study, two marker genes, GFP and GUS, were used to determine transformation efficiency. PCR amplification of the GUS gene in putative transgenic explants confirmed its presence. Our observations, including GFP fluorescence under the microscope, histochemical GUS staining (ranging from dark to light blue patterns in single or multiple cells), and PCR amplification, indicated that gene expression in alfalfa explants was not solely due to *Agrobacterium*, but also from transformed cells. This result confirmed the successful integration of the transient gene expression from pCAMBIA1304 into the alfalfa genome of Regen-SY. These analyses effectively minimized the occurrence of false positive transformants. Similar findings have been previously reported on the stable integration of the GUS gene, where infection with *Agrobacterium* significantly enhanced transformation efficiency. The transformed plants expressing the GUS gene was confirmed through PCR analysis in chickpea [46], Carrizo citrange [76], Matsumura [88], and Veratrum dahuricum [89].

## 4. Materials and Methods

### 4.1. Establishment of In Vitro Cultures

Seeds of the alfalfa cultivar Regen-SY germplasm (PI 537440) were collected from Western Regional PI Station through the U.S. National Plant Germplasm System. Alfalfa seeds were surface sterilized with 70% ethyl alcohol for 30 s followed by 20% bleach (Clorox^®^) treatment for 10 min [90]. The treated seeds were rinsed with sterile distilled water three-to-five times and then placed in 100 mm × 15 mm Petri dishes on Murashige and Skoog (MS) basal medium (PhytoTechnology Laboratories, Lenexa, KS, USA) containing 3% sucrose and 0.8% *w*/*v* agar at pH 5.7.

The Petri dishes were kept in the dark for 3 weeks at 24 °C. After 3 weeks, nine germinated seeds per Petri dish were transferred in the magenta GA7 boxes containing MS basal medium with 3% sucrose and 0.8% *w*/*v* agar and incubated at a temperature of 24 °C for 7 days in a growth chamber supplied with a 16/8 h photoperiod using cool, white, fluorescent light (75 lmol s^−1^ m^−2^). Emerged seedlings with well-developed shoots and roots were maintained in a growth chamber, and only healthy plantlets were chosen for further propagation. Sterile scalpels were used to excise 3-week-old subcultured plants with four different explants, namely, the leaves, stems, petioles, and roots. Callus was obtained from alfalfa leaf explants following the modified protocol [90,91,92]. Fully expanded leaves from 3-week-old alfalfa plants, were carefully removed, and cut into small segments of uniform size (≈2 mm). Leaf explants were placed on a callus induction medium using Gamborg’s B5 basal (B5H) medium, supplemented with 3% maltose, 4.5 μM 2,4-dichlorophenoxyacetic acid (2,4-D), 0.9 μM kinetin, 6.65 g of glutamine, 0.8 g of serine, 0.004 g of adenine, and 0.08 g of L-glutathione. The cultures were incubated in the dark at 24 ± 2 °C for 3 weeks [92,93].

### 4.2. Preparation of Agrobacterium Strain and Plasmid Vector for Transformation

*Agrobacterium tumefaciens* strain GV3101 harboring the pCAMBIA1304 plasmid expressing both *β-glucuronidase* (GUS) and green fluorescent protein (GFP) genes fusion was used for transformation. Both GFP-GUS gene fusions were under the control of a *Cauliflower mosaic virus* (CaMV) 35s promoter and NOS terminator. The T-DNA region of the plasmid also contained the hygromycin B phosphotransferase gene, driven by the CaMV 35s promoter, as a selection marker. The pCAMBIA1304 has been routinely used to demonstrate or establish the transient expression of several plant species, such as *Arabidopsis* [31], several medicinal plants [94], rice [95], tobacco [96], and *Dendrobium Sonia* [97]. The vector pCAMBIA1300—not containing reporter genes—was used as a control. A single *Agrobacterium* colony derived from stock culture as described by Dutt and Grosser (2009) [59] was cultured in liquid Luria–Bertani (LB) medium [94] containing 100 mg L^−1^ kanamycin and was grown on a shaker at 120 rpm at a temperature of 28 °C for 16 h.

### 4.3. Infection and Co-Cultivation of Alfalfa with A. tumefaciens

Glycerol stocks containing 12% glycerol were utilized to maintain the *Agrobacterium* culture at −20 °C, with initiation 3 days prior to weekly experiments. The *Agrobacterium* strain GV3101 with construct pCAMBIA1304 was grown on Luria–Bertani (LB) agar containing 100 mg/L kanamycin and 100 mg/L gentamicin at 28 °C for 2 days. A single colony of *Agrobacterium* from a fresh subculture plate was collected with a sterile loop inoculated into 2 mL of LB broth liquid medium containing 100 mg/L kanamycin and 100 mg/L gentamicin for 8 h at 28 °C, 200 rpm. The following day, 200 µL of bacterial suspension was cultured from the stock into 15 mL of liquid LB medium and grown overnight (24 h) at 28 °C in a shaker with 200 rpm agitation until the desired density at OD_600nm_ was obtained. The selected explants were then inoculated with the overnight grown *Agrobacterium*. To eliminate *Agrobacterium*, the explants were washed twice with MS basal liquid medium, dried using filter paper, and then placed on a selection medium containing Gamborg’s B5 basal (B5H) medium, supplemented with 3% maltose, 4.5 μM 2,4-dichlorophenoxyacetic acid (2,4-D), 0.9 μM kinetin, 6.65 g glutamine, 0.8 g serine, 0.004 g adenine, 0.08 g L-glutathione, 4 mM calcium chloride, and 75 µM silver nitrate (Figure 15).

### 4.4. Optimization of Transient Gene Expression Parameters

Several factors affecting *Agrobacterium*-mediated transformation frequency in alfalfa were evaluated. Factors such as explant type (leaf-, stem-, petiole-, root-, and leaf-derived 3-week-old calli), the age of the explant (0, 1, 3, 5, 7, and 9 weeks old), the size of leaf explants (sectioned and whole), wounding technique (control, unwounded, or intact; wounding with cell strainer pestle; wounding with 80 µm mash; wounding with a scalpel; wounding with a 0.1 mm needle), *Agrobacterium* concentration (0, 0.1, 0.2, 0.4, 0.6, 0.8, 1.0, and 1.2 at OD_600nm_), preculture period (0, 1, 2, 3, 4, 5, 6, and 7 days), infection period (0, 5, 10, 15, 20, 25, 30, and 35 min), co-cultivation period (0, 1, 2, 3, 4, 5, 6, and 7 days), acetosyringone concentration in infection solution (0, 50, 100, 150, 200, 250, 300, and 350 µM), calcium strength in culture medium (0, 1, 2, 3, 4, 5, 6, and 10 mM), and silver nitrate in culture medium (0, 10, 25, 50, 75, 100, 125, and 150 µM) were investigated.

Several parameters were used to determine the optimal conditions for transformation, and their effects on the percentage of GFP and transient GUS gene expression were evaluated. Explants co-cultured without agro-infection were used as controls. At the end of the co-cultivation period, the treated leaf explants were assessed and optimized based on their GFP gene expression (visible under a microscope with fluorescent green light) and GUS gene expression (visible as blue spots). The results were determined based on the percentage of explants exhibiting GFP and GUS spots out of the total number of inoculated explants, which were observed 3 days after transformation. A sample was considered GUS-positive if at least 25% of the leaf explant exhibited blue spots. This experiment was conducted three times, and equivalent results were obtained. The experiments were performed using 16 individual explants and repeated four times.

### 4.5. GFP Gene Detection, GUS Histochemical Assay, and Agrobacterium Infection Efficiency

GFP gene expression in alfalfa tissues was observed using an Olympus SZX12 Stereo fluorescence-equipped microscope with an HBO 100 W mercury bulb light mounted with a long-pass GFP filter supplied with a DP72 camera. The magnification level used for the observation was 16×. The GFP filter with excitation wavelengths of 460–480 nm, emission wavelengths of 495–540 nm, along with a GFPA filter for separation of GFP and blue-excited fluorescence with excitation wavelengths of 460–490 nm and emission wavelengths of 510–550 nm, was used to identity the leaves, stems, petioles, roots, and embryogenic calli expressing GFP events. Photographs were taken with an Olympus SZX12 automatic exposure photomicrographic system. To determine the percentage of transient GFP expression, the number of explants producing a GFP positive was divided by the total number of explants co-cultivated with *Agrobacterium*. An explant was considered positive for transient expression if it had ten or more cells expressing GFP.

Confirmation of transient transformed events for GUS activity was performed in different tissues according to the method described by Jefferson in 1987 [21]. After 3 days of co-cultivation, the alfalfa explants were immersed in a solution of 2 mM X-Gluc (5-bromo-4-chloro-3-indolyl—β-D-glucuronide) containing 0.1 M NaH2PO4 (pH 7.0), 0.1 M potassium ferricyanide, 0.1 M ferrocyanide, and 0.1% (*v*/*v*) Triton X-100. Before the observation of transient gene expression in different explants, they were incubated at 37 °C for 16–24 h. For removing chlorophyll, the X-Glue solution was replaced with double-distilled water followed by washing with 75% ethanol. The mixture of acetone/methanol (1:3) was added and incubated at 4 °C for 1 h. The explants were washed with double distilled water 4–5 times after obtaining clear tissue and then stored in 50% glycerol. The transient GUS activity of the alfalfa explants was examined under a Leica microsystems EMSPIRA 3 digital microscopes with a magnification of 1.2× (Life Sciences companies @ Danaher corporation, Washington, DC, USA) and scored as blue spots, irrespective of size.

The efficiency of *Agrobacterium* infection was assessed by calculating the percentage of transient GUS expression in alfalfa explants. The rate of transient GUS expression (%) was determined using the following formula: transient GUS expression rate (%) = (number of explants showing blue coloration/total number of stained explants) × 100%. Each experiment, aimed at optimizing individual parameters, included 16 samples and was replicated four times (*n* = 64).

### 4.6. Confirmation of Transgenic Samples via PCR Analysis

Before the PCR analysis, selected transgenic tissue segments were cultured in the LB medium for 36–48 h on 125 rpm in a shaker. By visual observation, no precipitation or change in color was observed in the LB medium, which indicates the absence of *Agrobacterium* in the transgenic tissue. To confirm, we checked the O.D. of the transgenic tissue growing liquid medium and control (LB medium) and found the same results.

Approximately 0.1 g alfalfa tissues of the putative transformants and non-transformed control were homogenized in liquid nitrogen using a mortar and pestle. Alfalfa genomic DNA was extracted following the manufacturer’s instructions using a Qiagen DNA extraction kit (USA) and quantified with a Nanodrop 2000 spectrophotometer (Thermo Fisher Scientific, Waltham, MA), followed by gel electrophoresis [98]. Polymerase chain reaction (PCR) was performed to confirm the stable integration of the GUS gene in the putative alfalfa transformants. Specific primers for the GUS gene were used: forward primer 5′-GTCCTGTAGAAACCCCAACCCGTGA-3′ and reverse primer 5′-CACTTCCTGATTATTGACCCACACT-3′, targeting a 493 bp fragment. Each PCR reaction was conducted in a 10 μL volume containing 10× standard reaction buffer, 5 mM dNTP, 0.5 µM of each primer, 50 ng of genomic DNA template, and 0.125 U of Taq DNA polymerase (New England Biolabs, Inc., Ipswich, MA, USA). The PCR cycling conditions consisted of an initial activation at 94 °C for 2 min, followed by 35 cycles of 94 °C for 30 s, 59 °C for 30 s (annealing), and 72 °C for 1 min, with a final extension at 72 °C for 5 min. The amplified PCR product was visualized on a 1.0% (*w*/*v*) agarose gel in Tris-acetate–EDTA (TAE) buffer and 0.1% SYBR ^TM^ Safe DNA gel stain (Thermo Fisher Scientific, Waltham, MA, USA) [99]. The band size was analyzed using a GelDoc system (Bio-Rad, Dublin, CA, USA) [46].

### 4.7. Statistical Analysis

Statistical analysis was conducted using one-way analysis of variance (ANOVA), and treatment means were compared using the Tukey–Kramer honestly significant difference (HSD) test at *p* ≤ 0.05 [100]. Graphs were generated using Prism v. 5.0 (GraphPad Software, La Jolla, CA, USA) and Microsoft Excel v. 2409.

## 5. Conclusions

Genetic transformation is a promising technique for improving crops through incorporating desired traits into their genomes. However, a significant challenge in this process is to increase the responsiveness of cells to *Agrobacterium*-mediated transformation. Hence, it is essential to identify and optimize critical parameters that positively influence *Agrobacterium*-mediated transformation efficiency during the infection and co-cultivation stages. This study aimed to improve existing protocols through systematically evaluating various parameters to achieve higher levels of transient gene expression in alfalfa. Additionally, this study extends the benefits of the *Agrobacterium*-mediated transformation system by employing a dual reporter system—GUS and GFP—along with hygromycin as a selectable marker. The GUS histochemical assay developed for alfalfa in this study could offer a rapid method for visualizing spatial gene expression and functionally testing gene promoters and other cis-regulatory elements in alfalfa. Furthermore, the GFP reporter gene is used as a form of non-invasive and real-time visualization, enabling the tracking of gene expression without damaging the cells and tissues. The dual reporter, combining GUS and GFP, enhances the precision of monitoring and visualizing gene expression and provides cross-validation of results by reducing the likelihood of false positives or negatives and validating the efficiency of gene expression.

Standardizing these parameters will facilitate the efficient transfer of agronomically important traits for the improvement of alfalfa. Moreover, optimizing transient gene expression in tissues may require the use of various additives to manage *Agrobacterium* overgrowth. Our research highlights key factors that influence T-DNA transfer efficiency and gene expression in alfalfa. Analysis of transient gene expression confirmed successful transformation in alfalfa using the *Agrobacterium* strain. The presence of blue spots in alfalfa explants indicates GUS reporter gene expression throughout the plants, driven by the constitutive CaMV 35s promoter. This successful *Agrobacterium-mediated* transformation of alfalfa could be further optimized for generating stable transformants using genome editing techniques such as CRISPR/Cas9, facilitating the functional testing of both genes and cis-regulatory regions such as promoters and enhancers. Our findings not only enhance the existing alfalfa transformation protocol but also emphasize the importance of a standardized transformation protocol to maximize the yield of transgenic alfalfa plants.

## Figures and Tables

**Figure 1 plants-13-02992-f001:**
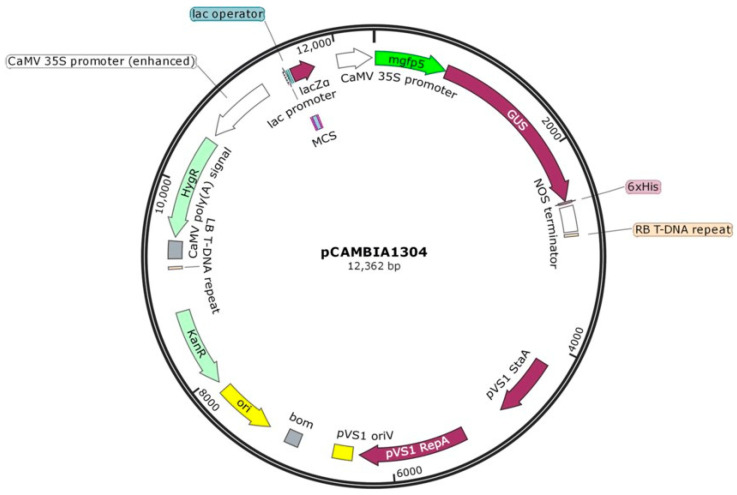
A schematic diagram of the plasmid vector pCAMBIA1304, represented within *Agrobacterium tumefaciens*, was used for the transformation of various alfalfa tissues. This vector includes several key elements: a kanamycin resistance gene (KanR) for bacterial selection, a hygromycin resistance gene (HygR) for plant cell selection, and an *mgfp5-GUS* fusion gene expression cassette, driven by the CaMV 35S promoter and terminated by the NOS poly-A sequence, serving as a reporter gene. The pCAMBIA1304 vector is relatively small, measuring 12,362 base pairs, and features a high copy number in *E. coli*, ensuring efficient DNA yields.

**Figure 2 plants-13-02992-f002:**
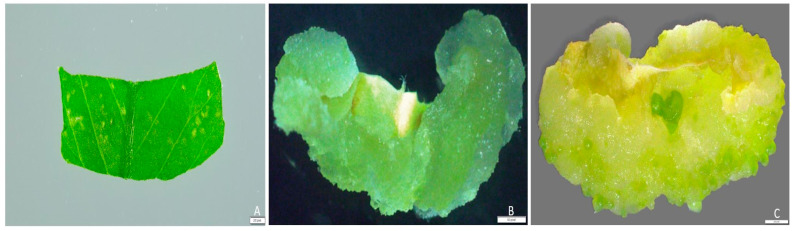
Different developmental stages of callus induction: (**A**) approximately 2 mm excised leaf segments cultured on callus induction medium; (**B**) 2-week-old callus derived from segmented leaf; (**C**) 3-week-old leaf-derived callus. Scale bars: 200 pixels (**A**, **C**); 50 pixels (**B**).

**Figure 3 plants-13-02992-f003:**
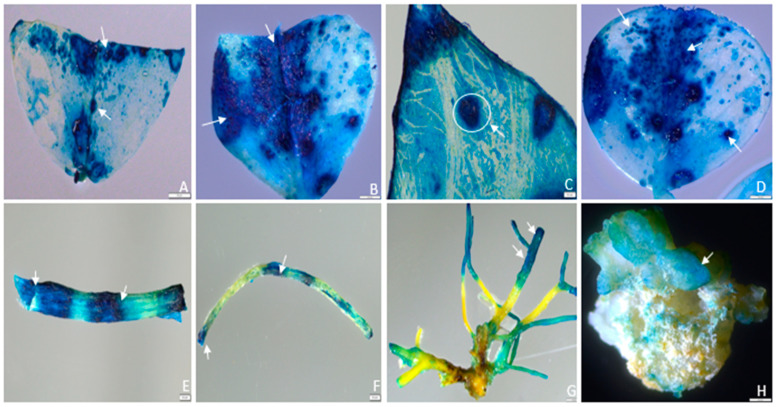
Histochemical localization of transient GUS gene expression in alfalfa different types of explants: (**A**) GUS gene expression in segmented leaf explants (0.2 cm), indicated by dark blue spots; (**B**) prominent GUS gene expression observed in the midrib region of a larger leaf explant (0.4 cm); (**C**) the circle shows a magnified view (6×) of a single blue spot with GUS expression in multiple cells within the midrib area; (**D**) dark-to-light blue pattern within single or multiple cells, indicating GUS-positive cells’ distribution across the leaflet; (**E**) active GUS expression in the axillary bud and nodal regions of the stem, marked by dark spots with enhanced expression; (**F**) subtle GUS gene expression observed in the petioles; (**G**) high levels of GUS gene expression observed in the lateral root area; (**H**) scattered GUS expression observed in 3-week-old leaf-derived callus. Arrows indicate specific regions of GUS gene expression in different explants and leaf-derived callus. Scale bars = 200 pixels.

**Figure 4 plants-13-02992-f004:**
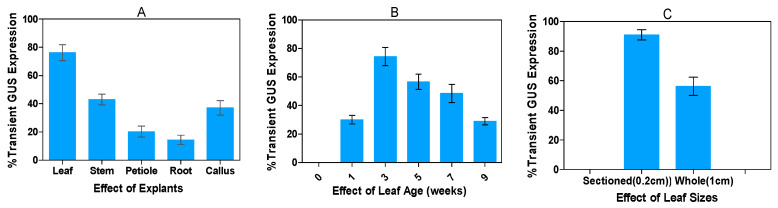
Optimization of transient GUS gene expression in alfalfa: Percent transient GUS gene expression response relative to the control alfalfa under varying rates of different parameters. (**A**) Effects of different explants on transient GUS gene expression. (**B**) Effect of age of leaf on transient GUS gene expression. (**C**) Effects of different sizes of leaf explants on transient GUS gene expression. All experiments were replicated four times (*n* = 64). Results were pooled over experimental runs. Vertical bars are represented as means ± standard errors.

**Figure 5 plants-13-02992-f005:**
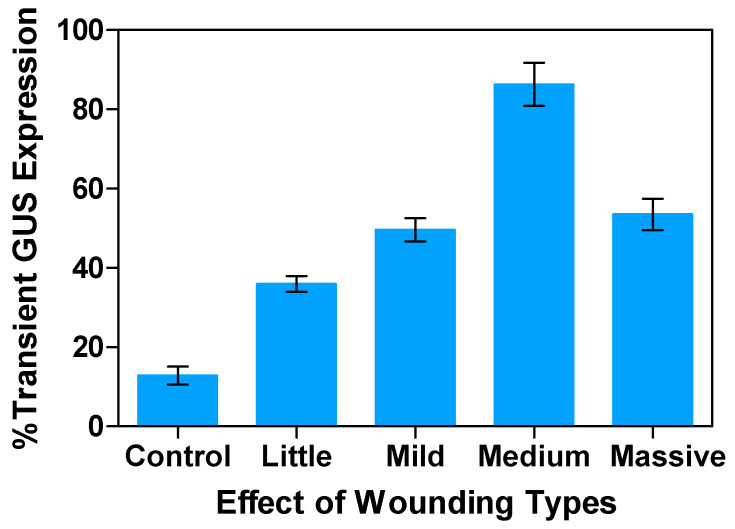
Effect of wounding types on transient GUS gene expression in alfalfa leaf explants: Percentage expression response in comparison to control under varying leaf tissue wounding conditions. All experiments were replicated four times (*n* = 64). Results were pooled over experimental runs. Vertical bars are represented as means ± standard errors.

**Figure 6 plants-13-02992-f006:**
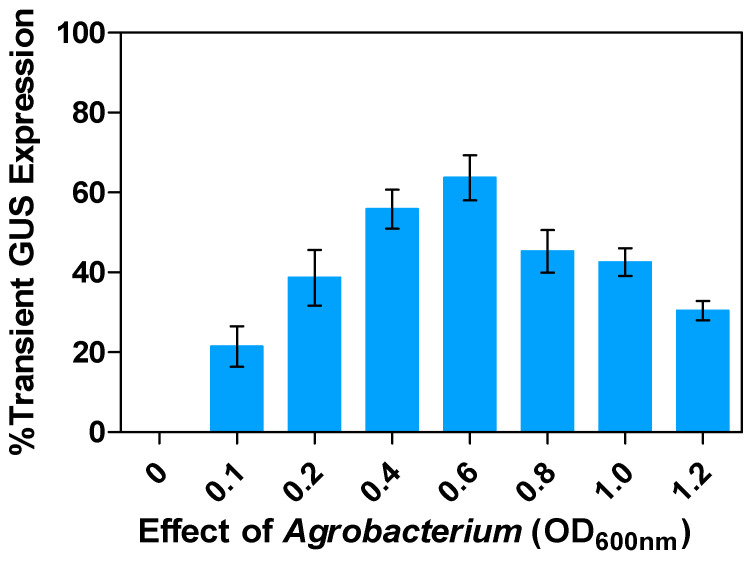
Effect of *Agrobacterium* concentrations on transient GUS gene expression in alfalfa leaf explants: Percentage transient response across varied effect of *Agrobacterium* concentrations compared to controls. All experiments were replicated four times (*n* = 64). Results were pooled over experimental runs. Vertical bars are represented as means ± standard errors.

**Figure 7 plants-13-02992-f007:**
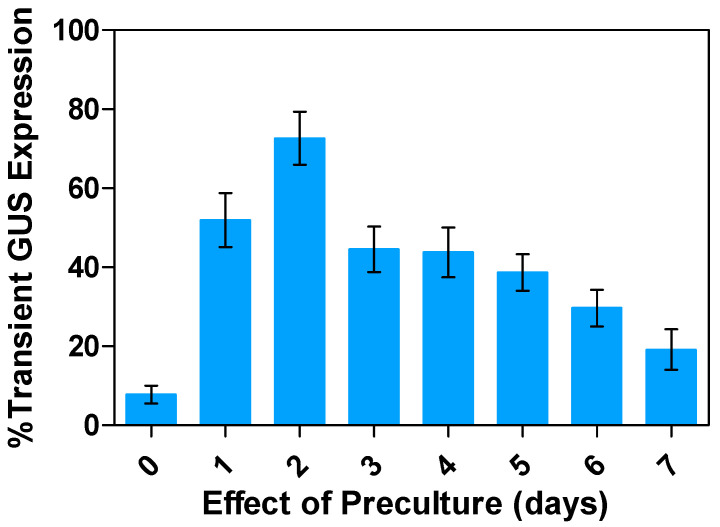
Impact of preculture of leaf tissues on transient GUS gene expression in alfalfa: Percentage expression relative to the control (in days). All experiments were replicated four times (n = 64). Results were pooled over experimental runs. Vertical bars are represented as means ± standard errors.

**Figure 8 plants-13-02992-f008:**
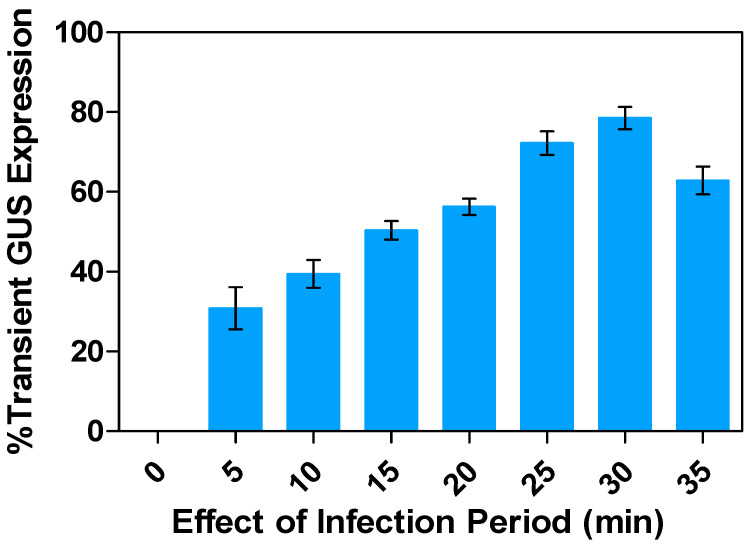
Influence of infection periods on transient GUS gene expression in alfalfa leaf explants: Percentage response rate compared to the control. All experiments were replicated four times (*n* = 64). Results were pooled over experimental runs. Vertical bars are represented as means ± standard errors.

**Figure 9 plants-13-02992-f009:**
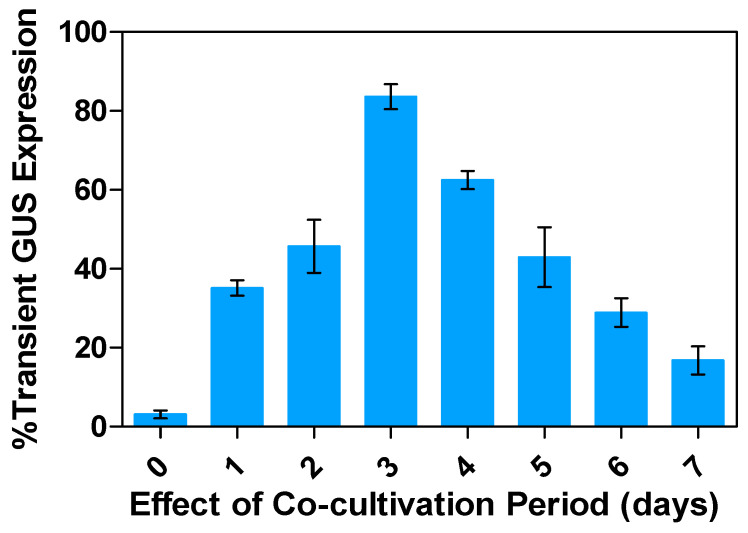
Evaluating transient GUS gene expression in alfalfa leaf explants: Percentage response under varying effect of co-cultivation period relative to the control. All experiments were replicated four times (*n* = 64). Results were pooled over experimental runs. Vertical bars are represented as means ± standard errors.

**Figure 10 plants-13-02992-f010:**
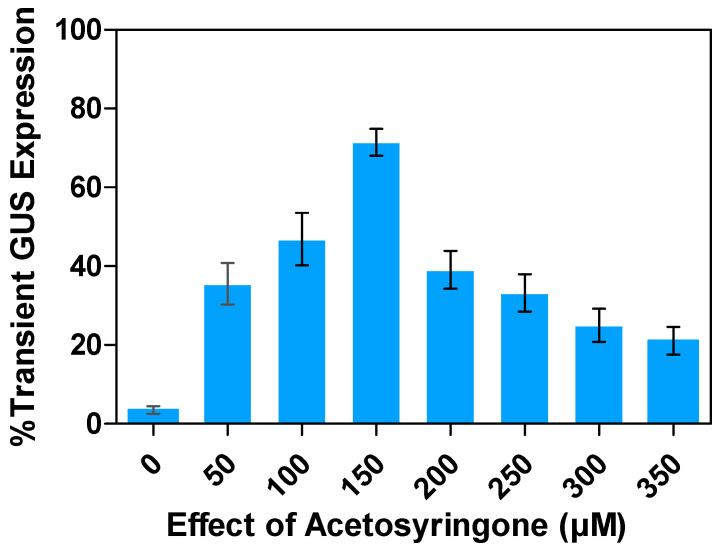
Effect of acetosyringone on transient GUS gene expression in alfalfa leaf explants: Percent expression compared to control plants. All experiments were replicated four times (*n* = 64). Results were pooled over experimental runs. Vertical bars are represented as means ± standard errors.

**Figure 11 plants-13-02992-f011:**
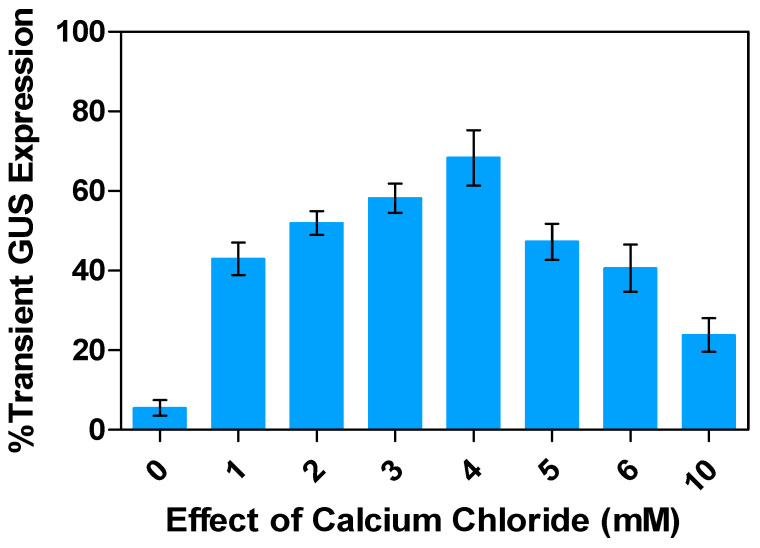
Effect of calcium chloride on transient GUS gene expression in alfalfa leaf explants: Percentage transient response under different calcium chloride concentration treatment relative to control. All experiments were replicated four times (*n* = 64). Results were pooled over experimental runs. Vertical bars are represented as means ± standard errors.

**Figure 12 plants-13-02992-f012:**
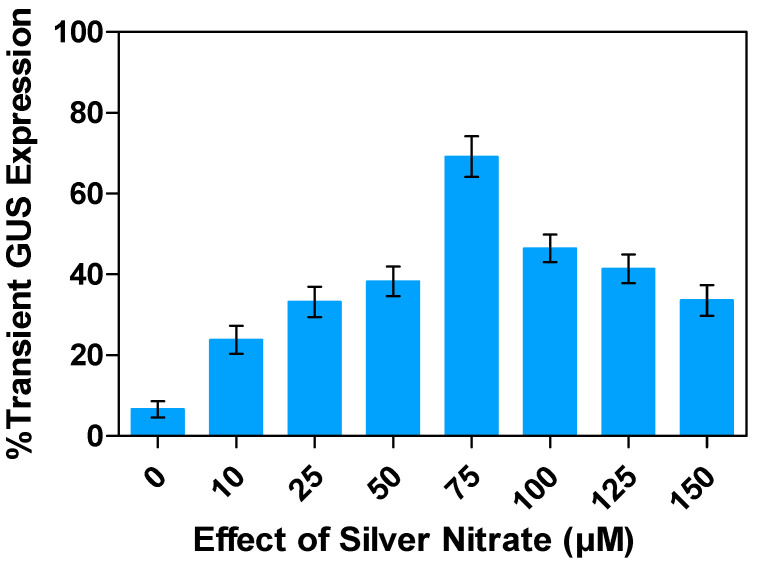
Assessment of the effects of silver nitrate on transient GUS gene expression in alfalfa leaf explants: Percent expression response compared to the control plants. All experiments were replicated four times (*n* = 64). Results were pooled over experimental runs. Vertical bars are represented as means ± standard errors.

**Figure 13 plants-13-02992-f013:**
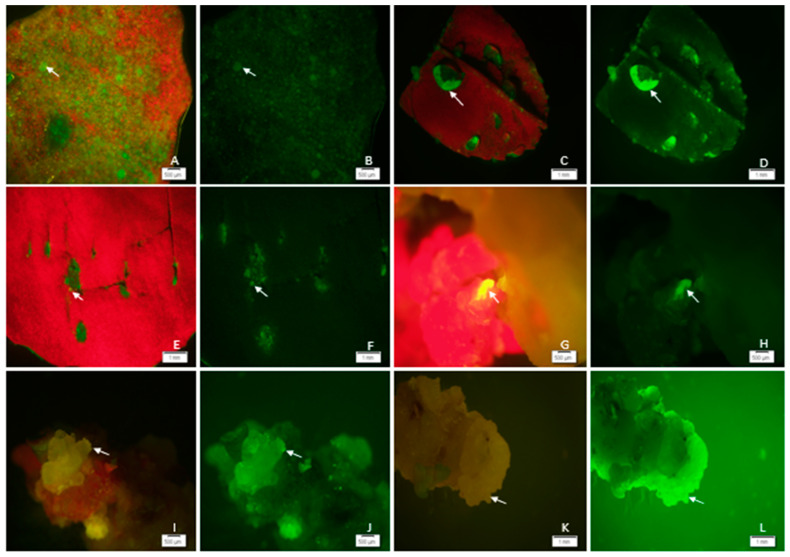
GFP fluorescence in alfalfa leaf explants (**A**–**F**) and embryonic calli (**G**–**L**): After co-cultivation with *Agrobacterium* at OD_600nm_ = 0.6 for 3 days, followed by two washes with liquid MS medium to eliminate *Agrobacterium*. (**A**) (chloroplast autofluorescence), (**B**) (GFP fluorescence): Fluorescence image of clustered multiple cells in 3-week-old wounded leaves (scale bar = 500 µm); (**C**) (chloroplast autofluorescence), (**D**) (GFP fluorescence): Isolated GFP spots in excised leaf explants (scale bar = 1 mm). (**E**) (chloroplast autofluorescence), (**F**) (GFP fluorescence): GFP expression in the wounded areas of leaf explants (scale bar = 1 mm). (**G**) (chloroplast autofluorescence), (**H**) (GFP fluorescence): GFP transient expression in 1-week-old pro-embryonic callus derived from leaves (scale bar = 500 µm). (**I**) (chloroplast autofluorescence), (**J**) (GFP fluorescence): GFP transient expression in 2-week-old pro-embryonic callus derived from leaves (scale bar = 500 µm). (**K**) (chloroplast autofluorescence), (**L**) (GFP fluorescence): GFP transient expression in 3-week-old pro-embryonic callus derived from leaves (scale bar = 1 mm). GFP expression was especially prominent in globular embryos from 3-week-old leaf-derived calli grown under non-selection conditions. Images were taken using an Olympus SZX12 Stereo Fluorescence Microscope with a GFP filter (Olympus America Inc., Melville, NY, USA). Arrows indicate GFP expression.

**Figure 14 plants-13-02992-f014:**
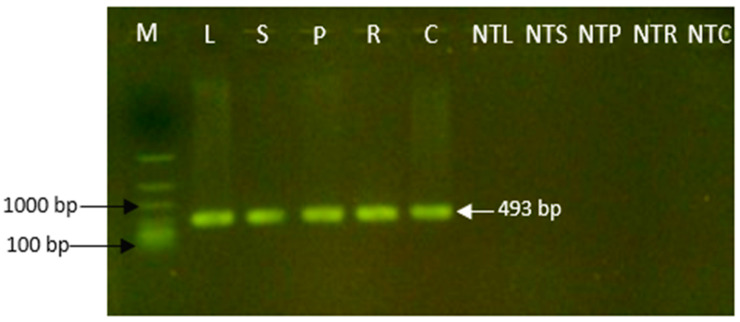
Molecular confirmation of alfalfa transient gene expression and analysis for the presence of the GUS (β-glucuronidase) gene in putative transgenic alfalfa explants using GUS-specific primers. Lane 1: M (marker DNA ladder, 100–4000 bp, FlashGEl^®^ DNA marker, Lonza Rockland, Inc., Rockland, ME, USA); Lanes 2–6: putative transgenic alfalfa explants, leaf (L), stem (S), petiole (P), root (R), and callus (C); Lanes 7–11: non-transgenic alfalfa explants, leaf (NTL), stem (NTS), petiole (NTP), root (NTR), and callus (NTC).

**Figure 15 plants-13-02992-f015:**
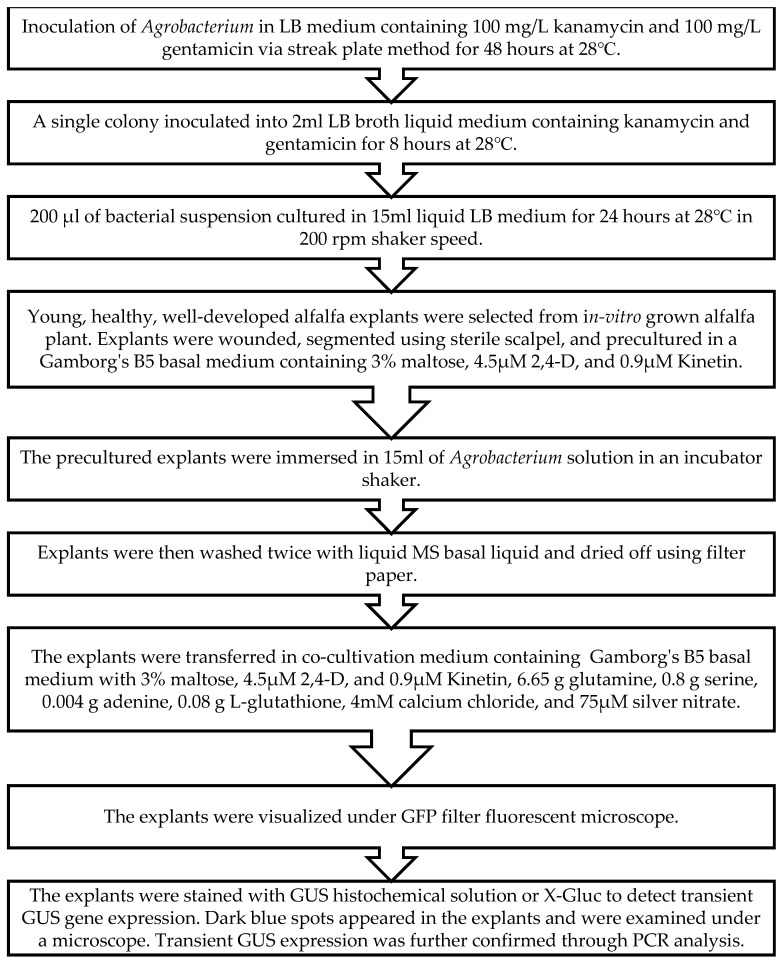
Step-by-step protocol for transient expression in *Medicago sativa* L. (alfalfa).

## Data Availability

All data are available within the manuscript, and further information can be obtained upon request from the corresponding authors.

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
